# Prevalence of *Pneumocystis jirovecii* Colonization in Non-Critical Immunocompetent COVID-19 Patients: A Single-Center Prospective Study (JiroCOVID Study)

**DOI:** 10.3390/microorganisms11122839

**Published:** 2023-11-22

**Authors:** Antonio Riccardo Buonomo, Giulio Viceconte, Ludovica Fusco, Marina Sarno, Isabella di Filippo, Luca Fanasca, Paola Salvatore, Ivan Gentile

**Affiliations:** 1Department of Clinical Medicine and Surgery, University of Naples “Federico II”, Via Sergio Pansini n.5, 8031 Naples, Italy; antonioriccardobuonomo@gmail.com (A.R.B.); ludo940f@gmail.com (L.F.); marinasarno92@gmail.com (M.S.); isadifi93@gmail.com (I.d.F.); ivan.gentile@gmail.com (I.G.); 2Department of Molecular Medicine and Medical Biotechnology, University of Naples “Federico II”, 8031 Naples, Italy; luca.fanasca@aulss3.veneto.it (L.F.); psalvato@unina.it (P.S.)

**Keywords:** COVID-19, *Pneumocystis jirovecii*, SARS-CoV-2, immunocompromised, pneumonia

## Abstract

Background: *Pneumocystis jirovecii* pneumonia (PJP) is an invasive fungal infection (IFI) that occurs mainly in immunocompromised hosts. After observing a high prevalence of PJP as a complication of COVID-19 in immunocompetent patients, we conducted a study to evaluate the prevalence of *P. jirovecii* colonization with PCR on oral washing samples (OWS) among non-immunocompromised and non-critical patients admitted with COVID-19 pneumonia at our university hospital. Methods: All patients over 18 years of age admitted to the Infectious Diseases Unit for SARS-CoV-2 pneumonia between July 2021 and December 2022 were included. Patients undergoing invasive mechanical ventilation or ECMO, those with risk factors for developing PJP, and those receiving prophylaxis for *P. jirovecii* were excluded. Samples were collected by gargling with 10 mL of 0.9% NaCl on day 14 of the hospital stay or at discharge. Results: Of 290 screened patients, 59 (20%) met the inclusion criteria and were enrolled. Only 1 of 59 patients (1.7%) tested positive for *P. jirovecii* detection with PCR, and the same patient was the only one to develop PJP in the follow-up period. Conclusions: Our results are in line with the previous findings of other studies that confirmed a very low prevalence of *P. jirovecii* colonization on OWS in the immunocompetent population. Despite the limitations of the study, the fact that the only patient who tested positive for *P. jirovecii* was the only one in our cohort to develop PJP leads us to reflect on the role of this non-invasive sample in predicting the risk of PJP in patients with COVID-19.

## 1. Background

*Pneumocystis jirovecii* pneumonia (PJP) is an invasive fungal infection (IFI) that occurs mostly in immunocompromised patients, especially in HIV-positive patients with a CD4+ lymphocyte count lower than 200 cells/mm^3^, solid organ transplant recipients and patients with hematologic malignancies or rheumatic conditions receiving prolonged doses of steroids or lymphocyte-depleting agents [1,2,3].

Despite being less frequent than invasive aspergillosis, PJP can complicate the course of COVID-19 in immunocompetent individuals as well, even though the exact prevalence of such IFI is not well established due to a lack of standardized criteria among published studies [4,5].

*P. jirovecii* can colonize the respiratory tract of asymptomatic individuals, especially when affected by chronic respiratory diseases, and can spread to non-colonized individuals via an airborne route in clinically evident diseases in case of impairment of the immune system of the host [6].

The definitive diagnosis of PJP can be made by detecting *P. jirovecii* on respiratory tract samples with direct immunofluorescence or traditional staining, or based on histopathological evidence, according to the Consensus Definitions of Invasive Fungal Disease From the European Organization for Research and Treatment of Cancer and the Mycoses Study Group Education and Research Consortium (EORTC/MSGERC) [7].

Polymerase chain reaction (PCR) for *P. jirovecii* on respiratory samples is a fast and sensitive alternative to direct microbiology. However, it cannot discriminate between infection and colonization; therefore, its positivity is considered a minor microbiological criterion, together with beta-D-glucan (BDG) serum detection, to diagnose probable PJP in patients with risk factors and clinical–radiological signs of PJP [7].

Although the preferred sampling technique to diagnose PJP is the bronchoalveolar lavage (BAL), non-invasive sampling of the respiratory tract with induced sputum, nasopharyngeal aspirate and oral washing samples (OWS) have shown high sensitivity, especially with the aim of detecting colonization through molecular analysis such as PCR [8,9]. In fact, several studies that aimed to describe the prevalence of *P. jirovecii* colonization have been carried out with the use of PCR on oral wash specimens and in healthy immunocompetent individuals, but never in COVID-19 patients [8,10].

Specifically, according to a 2021 meta-analysis by Senécal and colleagues, PCR on OWS had an overall pooled sensitivity of 77% (95%CI 66–85%) and specificity of 94% (95%CI 90–96%) among included studies, with a negative predictive value above 95% if the prevalence of PJP was below 17% and a positive predictive value above 90% if the prevalence was above 42% [9].

Data from a prospective study by Hviid and colleagues, which aimed to validate a quantitative real-time PCR assay for *P. jirovecii* on OWS in a population of asymptomatic HIV, transplant and rheumatologic patients, showed that Pneumocystis carriage was most frequent in renal transplant patients within the first 6 months after operation and in Danish HIV-infected patients, and less frequent among untreated HIV-infected patients from Guinea Bissau and among patients on immunosuppressive therapy [8]. In the study, a fungal load of >1000 copies was associated with symptomatic PJP, while a load <30 was associated with asymptomatic carriers, who did not develop symptomatic disease upon follow-up [8].

At our university hospital, we observed and reported an unexplainable high prevalence of PJP as a complication of COVID-19, even in patients who were immunocompetent before SARS-CoV-2 infection. On the other hand, other published studies reported PJP mostly in HIV or transplant individuals with COVID-19 [11,12,13]. Therefore, we conducted a study to evaluate the prevalence and features of *P. jirovecii* colonization with PCR on oral wash samples among non-immunocompromised and non-critical patients admitted with COVID-19 pneumonia at our institution.

## 2. Methods

### 2.1. Population

All the patients over 18 years of age affected by SARS-CoV-2 pneumonia and admitted to the Infectious Disease Unit of the Federico II University Hospital between 1 July 2021 and 31 December 2022 were screened for enrollment.

### 2.2. Exclusion Criteria

-Invasive mechanical ventilation or ECMO at enrollment.-Patients on *P. jirovecii* prophylaxis or who are chronically taking or who have taken active drugs against *P. jirovecii* (trimethoprim-sulfamethoxazole, pentamidine, atovaquone, dapsone) within the last month.-HIV infection.-Solid organ or hematogenous stem cell transplant recipients.-Active hematologic malignancies.-Connective tissue diseases with a history of prolonged steroid therapy (>20 mg of prednisone or equivalent for >3 weeks) and/or lymphocyte-depleting agents.-Previous diagnosis of PJP during their lifetime.-Patients unable to express consent to participate.-Patients unable to produce OWS.

### 2.3. Definitions

Colonization from *P. jirovecii* was defined as: (i) absence of signs and symptoms of PJP; (ii) respiratory specimen with detectable *P. jirovecii* DNA by PCR; (iii) no criteria fulfilled for definitive or probable PJP according to EORTC/MSGERC.

The diagnosis of PJP was considered according to the EORTC/MSGERC criteria: “proven” if *P. jirovecii* was detected with direct immunofluorescence assay (DFA) on respiratory samples; “probable” in the presence of host factors, clinical features and microbiological evidence; “possible” in the presence of host and clinical features with the absence of microbiological evidence [7].

PJP severity was classified according to the 1996 classification by Miller et al. as mild, moderate or severe based on clinical features, peripheral oxygen saturation, chest radiology and arterial oxygen tension (PaO_2_) on room air [14]. Cut-offs for arterial oxygen tension (PaO_2_) at rest in room air were: >11.0 kPa (>82.5 mmHg) for mild disease; 8.1–11.0 kPa (60.75–82.5 mmHg) for moderate disease and <8.0 kPa (<60 mmHg) for severe disease [14].

Charlson Comorbidity Index comorbidities were calculated to evaluate each patient’s comorbidities [15]. Steroid dose was calculated as being equivalent to dexamethasone since it was the most frequently used steroid drug in the studied population. Chronic steroidal treatment was defined as a daily dose >0.3 mg/kg of prednisone or equivalent for >2 weeks taken by the patient in the past 60 days, according to the EORTC/MSGERC criteria for host factors [7].

COVID-19 severity was assessed with the World Health Organization 9-point severity scale as follows: 0: no clinical or virological evidence of infection; 1: ambulatory, no activity limitation; 2: ambulatory, activity limitation; 3: hospitalized, no oxygen therapy; 4: hospitalized, oxygen mask or nasal prongs; 5: hospitalized, non-invasive mechanical ventilation (NIMV) or high-flow nasal cannula (HFNC); 6: hospitalized, intubation and invasive mechanical ventilation (IMV); 7: hospitalized, IMV + additional support such as pressors or extracardiac membranous oxygenation (ECMO); 8: death [16].

### 2.4. Sampling and Data Collection

As per routine clinical protocol at the Infectious Diseases ward, all the patients admitted for SARS-CoV-2 pneumonia undergo a full blood picture and complete biochemical blood and urine analysis, HIV antibody test with 5th-generation ELISA assay, chest X-ray and/or chest CT scan and arterial blood gas analysis.

Enrolled patients were asked to sign an informed consent form and to produce an OWS by gargling with 10 mL of sterile physiologic serum (0.9% NaCl) for a period of 2 min on the 14th day of hospital stay or at discharge, whichever came first. A serum sample for BDG detection was collected on the same day as the oral wash sample.

Included patients were followed up for a period of 3 months with monthly scheduled visits at the post-COVID-19 outpatient clinic of our institution and they were provided with a 24 h telephone contact number for the study center, to use to report any symptoms or worsening of their status. Patients requiring medical attention were seen at the outpatient clinic within 48 h from the call.

OWS were stored at −80 °C after collection and only analyzed after the end of the follow-up study.

### 2.5. Laboratory Analysis

The DNA of *P. jirovecii* was analyzed in oral wash samples using real-time PCR. The specimens were equilibrated at room temperature (RT) and 2 mL of samples was centrifuged for 10 min at 10,000 rpm at RT and suspended in 190 µL of Buffer G2 (EZ1 DNA Tissue Kit, QIAGEN GmbH, Hilden, Germany) and in 10 µL of Proteinase K (EZ1 DNA Tissue Kit, QIAGEN GmbH, Hilden, Germany). The samples were incubated at 56 °C in a thermostatic bath (HAAKE Shaking Water Bath SWB25, Truganina, Victoria 3029, Australia) for 30 min and then the DNA was extracted from the specimens using the instrument EZ1 Advanced XL (QIAGEN GmbH), according to the manufacturer’s instructions. The detection and quantification of DNA of *P. jirovecii* was performed with the RealStar^®^ *Pneumocystis jirovecii* PCR Kit 1.0 (Altona Diagnostics GmbH, Hamburg, Germany), an in vitro diagnostic test based on real-time PCR technology. The whole process was monitored by adding 5 µL of Internal Control (IC) (Altona Diagnostics GmbH, Hamburg, Germany) to each sample before the DNA extraction, to confirm the nucleic acid extraction and to exclude PCR inhibition. The real-time PCR tests were performed according to the manufacturer’s protocol. Briefly, the amplification was carried out in a CFX96 Real-Time thermocycler (Bio-Rad, Hercules, CA, USA). Each PCR was performed with 10 µL of extracted DNA in a final reaction volume of 30 µL. The thermal cycling conditions consisted of a denaturation at 95 °C for 2 min, followed by 45 cycles of alternating incubations: denaturation at 95 °C for 15 s, annealing at 58 °C for 45 s and extension at 72 °C for 15 s. Negative and positive controls, provided in the kit, were included in each assay. The final results were analyzed using the CFX96 Real-Time fluorescence quantitative PCR software, version 2.3, May 2022 (Bio-Rad, Hercules, CA, USA). The samples were positive if there was a detection of the IC in the JOE^TM^ detection channel and of *P. jirovecii* DNA in the FAM ^TM^ detection channel. For positive samples, a quantification standards curve containing standardized concentrations of *P. jirovecii* specific DNA (Altona Diagnostics GmbH, Hamburg, Germany) was used to determine the concentration of *P. jirovecii* specific DNA. The presence of human DNA in the oral wash samples was assessed through a dye-based qPCR targeting the human beta-globin gene using the SsoFast EvaGreen Supermix kit. The primers and thermal cycling conditions can be provided if needed. This qPCR was executed on the same extracted DNA and was later used for the Pneumocystis qPCR; all the samples tested positive.

### 2.6. Statistical Analysis

The statistical analysis was performed using SPSS version 27 (SPSS Inc., Chicago, IL, USA). Continuous variables were reported as median and interquartile range and categorical variables as frequency and percentages. Categorial variables were analyzed using the Chi-squared test and Fisher’s exact test when appropriate. Continuous variables were analyzed using the Mann–hitney U test. A significance level of 0.05 was set for the interpretation of the results.

## 3. Results

We screened all the patients hospitalized for COVID-19 between July 2021 and December 2022 according to the inclusion criteria. Of the 290 patients screened, 59 (20%) met the inclusion and exclusion criteria to be enrolled. The median age of the study population was 62 years (IQR 35–68), with 61% female patients and 11 out of 59 (18.6%) being pregnant women. Only four (6.8%) patients had an intensive care unit admission in the 20 days before the enrollment. The most commonly reported comorbidity was chronic lung disease (17%), followed by chronic kidney disease (13.6%). Most of the patients required low-flow oxygen support with a Venturi mask (37.3%) or nasal cannula (25.4%) and, therefore, the median highest WHO grading was 4 (IQR 4.5). Detailed demographic and clinical characteristics of the patients are displayed in Table 1.

In the analysis, we also considered seven patients with a history of solid tumor in the past 5 years, as none of them had received chemotherapy within the last year. None of these patients received chemotherapy during the follow-up study.

Out of 59 oral washing samples collected on a median of 13 (IQR 9–13) days from admission, only 1 (1.7%) was positive for *P. jirovecii* genome detection with PCR, with a value of 500 copies, and the same patient was the only one to develop clinically evident *P. jirovecii* pneumonia 10 days after hospital discharge. The patient, an obese 77-year-old man affected by hypertension, was vaccinated for SARS-CoV-2 with two doses and was admitted at our institution with COVID-19 pneumonia 22 days before PJP diagnosis. He was treated with intravenous remdesivir and dexamethasone at 6 mg daily for 5 days and he received oxygen with a Venturi mask, with the highest need for FiO2 of 60%. He was discharged 12 days later on room air and in good condition with a negative nasopharyngeal swab for SARS-CoV-2 detection. Then, 10 days after hospital admission, he arrived at the outpatient clinic with fever, exertional dyspnea, coughing and a peripheral oxygen saturation of 89 on room air. A CT scan performed at that time showed bilateral diffuse ground glass opacities (Figure 1). Bronchoalveolar lavage fluid (BALF) was collected, which resulted in a positive detection of *P. jirovecii* via direct immunofluorescence, while other tests on BALF and serum/urine for respiratory viruses, bacteria and fungi, including mycobacteria, were negative. He was treated with intravenous trimethoprim-sulfamethoxazole (TMP-SMX) at the dose of 15 mg/kg divided into four daily doses for 3 days (with switch to oral therapy after reaching clinical improvement) plus prednisone at 40 mg twice daily for the first 5 days, followed by 40 mg daily for 5 days and 20 mg daily for the remaining 11 days. He was discharged 5 days after starting therapy for PJP in good clinical condition and with no oxygen requirement.

No other patients developed any signs or symptoms of PJP in the follow-up period. No patients, including the one who developed PJP, had detectable beta-D-glucan in serum on the day of collection of the oral washing sample or at the time of PJP diagnosis.

## 4. Discussion

We found a low (1.7%) prevalence of *P. jirovecii* colonization detected with OWS in a cohort of immunocompetent patients affected by COVID-19. Nonetheless, the only patient in our cohort who was affected by *P. jirovecii* was the only one who developed clinically significant PJP during follow-up. He had a value of 500 copies of *P. jirovecii* DNA on the OWS collected at hospital discharge and developed moderate PJP diagnosed with immunofluorescence staining on BAL.

According to the largest review available on the topic, published in 2021 by Vera C. and Rueda Z.V., the prevalence of *P. jirovecii* colonization detected with PCR on various respiratory samples (including OWS) is extremely variable across the included studies, ranging from 0% in healthy non-pregnant women to 50% in immunocompetent pregnant women and 70% in newborns and in patients with chronic obstructive pulmonary disease (COPD) or HIV infection [6].

On the other hand, according to a review by Morris et al., among seven published studies specifically aiming to search for *P. jirovecii* colonization in healthy immunocompetent hosts, five of them found a prevalence of 0% with PCR on several respiratory specimens, including BAL fluid and lung specimens from autopsies, on a number of subjects ranging from 10 to 30 [17]. Conversely, one study published in 1997 by Nevez and colleagues found 20% positive *P. jirovecii* PCR among 169 patients who underwent BAL for any reason (the largest cohort among the seven studies), while another paper in 2005 by Medrano and colleagues found the same prevalence with PCR on OWS for 50 healthy workers at a Spanish hospital with no underlying lung conditions [10,18].

Such differences in colonization prevalence in immunocompetent hosts seem related to different risk factors in the included populations, specifically hospitalized patients in the case of Nevez et al., and healthcare workers in the case of Medrano et al. [10,18]. Transmission of *P. jirovecii* from the hospital environment or from other colonized or infected patients is in fact a known route of colonization in immunocompetent hosts as well [6].

Thus, we decided not to test our study population at hospital admission, but after 14 days of hospital stay or at hospital discharge, whichever occurred first, since we expected a very low prevalence of *P. jirovecii* colonization in immunocompetent patients. Nonetheless, we still found that the prevalence of colonization from *P. jirovecii* on OWS in non-critical immunocompetent patients with COVID-19 was very low. On the other hand, despite the small number of enrolled patients, the fact that the only patient who tested positive for *P. jirovecii* on OWS was the same one who developed clinically significant PJP leads us to reflect about the role of this non-invasive sample in predicting the risk of PJP in COVID-19 patients.

According to EORTC/MSGERC criteria, a positive PCR for *P. jirovecii* on a respiratory sample, together with the presence of a host factor and a typical clinical–radiological picture, is sufficient for a “probable” diagnosis of PJP [7]. Less is known about the role of PCR in the preclinical context to stratify at-risk patients who may benefit from a close follow-up or prophylaxis, especially in atypical populations, such as COVID-19, where classic risk factors for PJP can be absent and the clinical–radiological picture is very similar to SARS-CoV-2 pneumonia.

Moreover, it is still debated whether the *P. jirovecii* PCR load can discriminate between colonization and infection, using a cut-off of 1000 copies/mL, according to several published studies [8,19]. In fact, according to Hviid and colleagues, a fungal load >1000 copies is associated with symptomatic PJP, while a load <30 is associated with asymptomatic carriage, but fungal loads within this interval fall into a “grey zone” that must be further explored [8].

In light of that, we propose that PCR for *P. jirovecii* on OWS should be interpreted with caution and according to the clinical scenario: In patients with clinically resolved COVID-19 pneumonia who do not develop novel symptoms of respiratory infections, the detection of *P. jirovecii* on OWS should be considered as colonization, regardless of fungal load; on the other hand, a positive PCR on OWS in symptomatic patients, especially with loads >1000, can be interpreted as a possible PJP that requires confirmation with a direct microbiological test (immunofluorescence on BAL).

A few questions remain unanswered: Which patients should be screened for *P. jirovecii* colonization? If colonized, should they undergo a closer follow-up? Which are the colonized patients who might require chemoprophylaxis for PJP?

## 5. Conclusions

Despite the limitations of our study, our results indicate that the prevalence of *P. jirovecii* colonization in OWS among immunocompetent, non-critical COVID-19 patients is very low, but seems to predict the development of clinically significant PJP. We can speculate that PCR on oral washing samples, other than the well-established diagnostic role in defining probable PJP, can have a role in the early identification of patients at risk of developing clinically significant PJP during the course of COVID-19, in addition to contributing to the selection of patients who might benefit from invasive diagnostics (bronchoscopy) and PJP chemoprophylaxis. Nonetheless, further studies on larger populations are required to evaluate the predictive value of this test on the COVID-19 population.

## Figures and Tables

**Figure 1 microorganisms-11-02839-f001:**
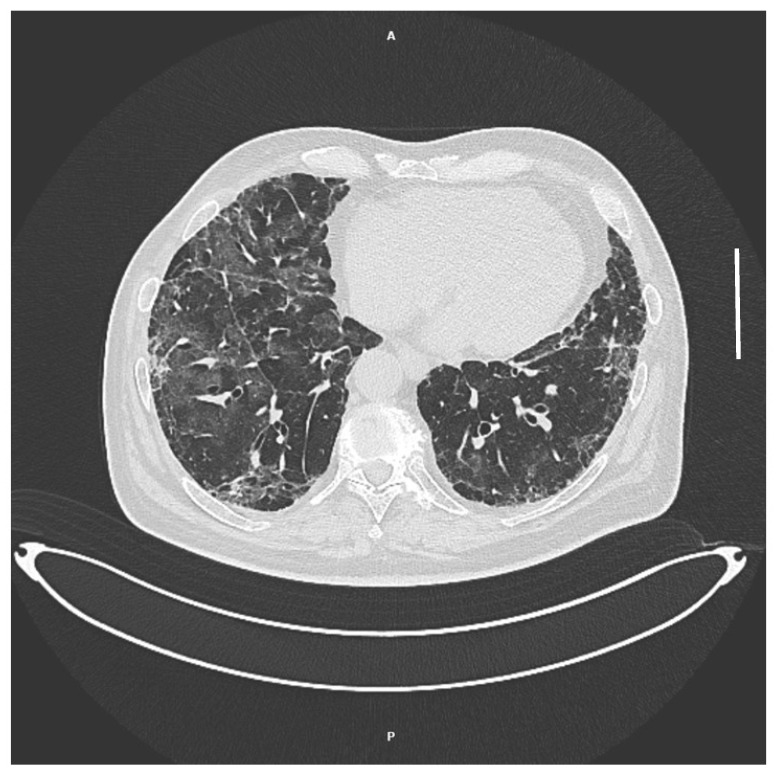
Patient’s chest CT scan at PJP diagnosis. A: anterior; P: posterior. Scale bar = 5 cm.

**Table 1 microorganisms-11-02839-t001:** Demographic and clinical characteristics of the population.

	N = 59
Age, years, median (IQR)	62 (35–68)
Females, n (%)	36 (61)
Pregnancy, n (%)	11 (18.6)
Length of stay, days, median (IQR)	16 (11–22)
Days from admission to sampling, median (IQR)	13 (9–13)
Admitted to ICU in the last 20 days, n (%)	4 (6.8)
Myocardial infarction, n (%)	6 (10.2)
Congestive heart failure, n (%)	4 (6.8)
Peripheral vascular disease, n (%)	2 (3.4)
Cerebrovascular disease, n (%)	5 (8.5)
Dementia, n (%)	0 (0)
Chronic lung disease, n (%)	10 (17)
Connective tissue disease, n (%)	1 (1.7)
Peptic ulcer, n (%)	1 (1.7)
Diabetes with organ damage, n (%)	1 (1.7)
Moderate/severe kidney disease, n (%)	8 (13.6)
Hemiplegia, n (%)	0 (0)
Moderate/severe liver disease, n (%)	2 (3.4)
Solid tumor in the last 5 years, n (%)	7 (12)
Charlson Comorbidity Index, median (IQR)	2 (0–6)
Days on steroid therapy, median (IQR)	13.5 (10–16.25)
Cumulative dose of steroid, mg, median (IQR)	70 (50–86)
Worst WHO grade, median (IQR)	4 (4–5)
Lowest PaO2/FiO2 ratio, median (IQR)	180.5 (130.25–269.75)
Days on highest oxygen support, median (IQR)	5 (5–8.5)
Lowest lymphocyte count cells/mm^3^, median (IQR)	655 (432–950)
Ferritin on admission, ng/mL, median (IQR)	204 (114–583)
CRP on admission, mg/dL, median (IQR)	5.5 (2.1–12.8)
Highest oxygen support required	
Nasal cannula, n (%)	15 (25.4)
Venturi mask, n (%)	22 (37.3)
CPAP, n (%)	3 (5)
HFNC, n (%)	9 (15.3)
NIV, n (%)	4 (6.8)

N: total number of cases; ICU: intensive care unit; WHO: World Health Organization; CRP: C-reactive protein; CPAP: continuous positive airway pressure; HFNC: high-flow nasal cannula; NIV: non-invasive ventilation.

## Data Availability

The data that support the findings of this study are available from the Federico II University Hospital, but restrictions apply to the availability of these data, which were used under license for the current study and so are not publicly available. Data are, however, available from the authors upon reasonable request and with permission of the Federico II University Hospital.

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
