# Peer review of "Prevalence of Pneumocystis jirovecii Colonization in Non-Critical Immunocompetent COVID-19 Patients: A Single-Center Prospective Study (JiroCOVID Study)"

_microorganisms, 2023, doi:10.3390/microorganisms11122839_

Round 1

Reviewer 1 Report (Previous Reviewer 4)

Comments and Suggestions for Authors

I think you have made all the corrections requested by reviewers

Author Response

thank you

Reviewer 2 Report (Previous Reviewer 3)

Comments and Suggestions for Authors

The authors have provided more information on sample quality control.

Author Response

thank you

Reviewer 3 Report (Previous Reviewer 2)

Comments and Suggestions for Authors

I want to thank you for considering my suggestions. I could notice the changes in the manuscript.

I have a few comments:

The font used in lines 45–47 is different from the one used in the rest of the manuscript.

Line 82: There is a period that is unnecessary.

Lines 39, 151, 2009, and 211: in scientific names, the genus name can be abbreviated after its initial use, and a full genus name is unnecessary.

 P. jirovecci shall be italicized in lines 106 and 107 and in the references.

In the text, reference numbers should be placed before the punctuation.

Author Response

  • line 45-47 the font has been uniformed
  • the period has been deleted
  • Names have been abbreviated as suggested
  • P. jirovecii has been italicized
  • Reference numbers have been placed before the punctuation.

This manuscript is a resubmission of an earlier submission. The following is a list of the peer review reports and author responses from that submission.

Round 1

Reviewer 1 Report

Comments and Suggestions for Authors

The authors aimed to evaluate the prevalence of Pneumocystis cause pneumonia in Covid-19 patients. 

Pneumocystis infection is quite common among immunosuppressed patients, but rare among immunocompetent ones. The authors mentioned an increased number of PCP cases among immunocompetent covid patients, but after excluding patients undergoing mechanical ventilation (which are much more prone to respiratory infections, anyway), only one positive case remained. Severe Covid-19 has already been shown to seriously compromise CD8+ immune response and innate mechanisms, which are teh most important for Pneumocystis response, so I'd hardly call these individuals immunocompetent, meaning there is no real surge of PCP in immunocompetent individuals. Other than that, the sole case is a 77-year old man, with hypertension and obesity, not a prime example of health and immunity either.

As for the main conclusion regarding the validity of the exam in oral wash, it would be an interesting one if it wasn't based off a sigle patient. It is an interesting thought and worth pursuing, as its a much easier sample to work with, but more data is required. 

Author Response

Reviewer.”The authors aimed to evaluate the prevalence of Pneumocystis cause pneumonia in Covid-19 patients. 

Pneumocystis infection is quite common among immunosuppressed patients, but rare among immunocompetent ones. The authors mentioned an increased number of PCP cases among immunocompetent covid patients, but after excluding patients undergoing mechanical ventilation (which are much more prone to respiratory infections, anyway), only one positive case remained. Severe Covid-19 has already been shown to seriously compromise CD8+ immune response and innate mechanisms, which are teh most important for Pneumocystis response, so I'd hardly call these individuals immunocompetent, meaning there is no real surge of PCP in immunocompetent individuals. Other than that, the sole case is a 77-year old man, with hypertension and obesity, not a prime example of health and immunity either.”

Reply:

We really appreciated reviewers’ comment. We want to take to opportunity to specify that the high number of PJP in immunocompetents we describe in the introduction is not referred to the results of the present study, but of our previous studies cited in the sentence. We also better specified in the same sentence (lines 74-75) that we consider patient to be immunocompetent before the onset of COVID-19 and its complications (including the possible lymphopenia that they might develop). As for lymphopenia, in another study that we cited in the introduction, we demonstrated that lymphopenia, as well as other classic risk factors for PJP (SOT, haematologic malingnacies) were not independent risk factors for PJP, this association in the above-metioned study has been demonostrated only for the cumulative dosage of corticosteroids administered for COVID-19  (reference n. 14)

Of course, immunosenescence and obesity are associated with decreased immune function, but since they are not typical risk factors for PJP, we considered our patient as immunocompetent.

Reviwer: As for the main conclusion regarding the validity of the exam in oral wash, it would be an interesting one if it wasn't based off a sigle patient. It is an interesting thought and worth pursuing, as its a much easier sample to work with, but more data is required. 

Reply:

We strongly agree with the reviewer. In fact, as we stated in our manuscript, further studies with larger populations are needed to better investigate the preliminary results of this manuscript and we will work in this way soon.

Reviewer 2 Report

Comments and Suggestions for Authors

I wish to express my sincere gratitude for the kind invitation to review this manuscript. The authors conducted a study to assess the prevalence of Pneumocystis jirovecii infection in non-critically ill, immunocompetent COVID-19 patients admitted to the University Hospital “Federico II”. P. jirovecii is a significant fungal pathogen primarily associated with immunocompromised individuals. Despite various reported prevalence rates, the true extent of P. jirovecii infections remains uncertain. Identifying the associated risk factors is crucial for advancing our understanding of this pathogen."

I just have a few comments:

Lines 81 – 84 should be removed since the information was already mentioned in the background (lines 76-79)

The Methods section shall encompass a description of the study's time frame.

The aim of the study was to determine the prevalence of P. jirovecii in non-critical, immunocompetent patients. However, it's important to note that the study included 7 patients with solid tumors and 2 with metastatic tumors. If these patients were undergoing intensive cancer chemotherapy, they would not meet the criteria for immunocompetent patients, and it is recommended that they be excluded from the study.

Infection should be defined in subtitle 2.3.

P. jirovecci shall be always italicized.

Author Response

We thank the reviewer for the comments, that we really appreciated. We reply accordingly as follow:

  • Lines 81-84 have been removed according to reviewer’s comment
  • A time frame was specified in lines 86-88
  • As for the 7 patients with history of solid tumor in the past 5 years, all of them were in complete remission and have not received chemotherapy within the last year, while the 2 patients with metastatic tumors were diagnosed with metastatic neoplasm during the hospitalization (one patients had a new diagnosis of colonic cancer with hepatic metastases and another of prostate cancer with bone metastases) and have never received any chemotherapy before the enrollment. This was reported in the result section (lines 212-217)
  • We defined in the first lines of the paragraph both the colonization from Pneumocystis jirovecii and the clinical diseases (Pneumocystis jirovecii pneumonia)
  • jirovecii has been italicized in the whole document

Reviewer 3 Report

Comments and Suggestions for Authors

The manuscript “Prevalence of Pneumocystis jirovecii colonization in non-critical immunocompetent COVID-19 patients: a single center prospective study” by A.R. Buonomo and colleagues, describes the importance of the prevalence of Pneumocystis jirovecii colonization among immunocompetent and non-critical COVID-19 adult patients and how it could be utilized to predict development of clinically significant PCP. The P. jirovecii colonization was assessed on 59 patients’ oral washing samples by real-time PCR. Despite only one patient having positive PCR for P. jirovecii, it was also the only patient to develop clinically evident PCP. This article is an important contribution to the development of P. jirovecii early detection tools and prevalence among immunocompetent patients. However there are some technical controls that need to be addressed.

1.     It is unclear if DNA quality was assessed or a human housekeeping gene was assayed which would be important to exclude false negatives. 

2.     Also the assay used targets mtLSU which may not has be abundant in asci and is subject to copy number variation: see https://www.ncbi.nlm.nih.gov/pmc/articles/PMC5018473/.  The authors should verify there results with mtSSU.  

Author Response

With regards to the first question, as specified in the text, in order to exclude false negatives, an Internal Control sample (IC) was added to the primary samples following the manufacturer's instructions. This extrinsic heterologous internal control was included in the kit and acted both as an extraction and an inhibition control.

Regarding the second question, the PCR kit targeting mtLSU was chosen based on the literature describing the diagnosis of P. jirovecii from oral wash samples (as an example, Hviid CJ et al., 2017 - Detection of Pneumocystis jirovecii in oral wash from immunosuppressed patients as a diagnostic tool). Furthermore, at the time of writing, as far as we could know, there were no available CE-IVD marked commercial kits targeting P. jirovecii mtSSU. 

Reviewer 4 Report

Comments and Suggestions for Authors

Dear authors:

This is an interesting article that shows a very low prevalence of P. jirovecii colonization among immunocompetent COVID-19 patients.

I only found a few mistakes:

P. jirovecii should be always written in italics.

There are two ways to name Pneumocystis jirovecii pneumonia: PJP (I like this one) and the old form PCP, please unify.

Best regards

Author Response

We thank the reviewer for the comments, that we really appreciated. We reply accordingly as follow:

  • The abbreviations have been checked and PCP was converted in PJP in the whole document.
  • jirovecii has been italicized in the whole document

Round 2

Reviewer 3 Report

Comments and Suggestions for Authors

The authors failed to make any revisions to the paper so the revised paper is essentially non responsive.  The authors refer to a spike in control but this would only control for post sample collection procedures.  To exclude a significant false negative rate, one would need to include a human housekeeping gene to assess adequate DNA extraction.    This is the problem with relying solely on a commercial assay for a research study. 

Author Response

We apologize for not have satisfied the reviewer's comment in the previous revision. We then specified in the methods section (lines 175-180) the technique we used to assess the presence of human DNA in the samples.